# Gut Microbial Shifts Indicate Melanoma Presence and Bacterial Interactions in a Murine Model

**DOI:** 10.3390/diagnostics12040958

**Published:** 2022-04-12

**Authors:** Marco Rossi, Salvatore M. Aspromonte, Frederick J. Kohlhapp, Jenna H. Newman, Alex Lemenze, Russell J. Pepe, Samuel M. DeFina, Nora L. Herzog, Robert Donnelly, Timothy M. Kuzel, Jochen Reiser, Jose A. Guevara-Patino, Andrew Zloza

**Affiliations:** 1Rush University Medical Center, Chicago, IL 60612, USA; marco_rossi@rush.edu (M.R.); timothy_kuzel@rush.edu (T.M.K.); jochen_reiser@rush.edu (J.R.); 2Rutgers Robert Wood Johnson Medical School, Rutgers, The State University of New Jersey, New Brunswick, NJ 08901, USA; saa9216@nyp.org (S.M.A.); rjpepe19@gmail.com (R.J.P.); 3Rutgers Cancer Institute of New Jersey, Rutgers, The State University of New Jersey, New Brunswick, NJ 08901, USA; kohlhapp@gmail.com (F.J.K.); jenna.newman@mssm.edu (J.H.N.); sam.defina@yale.edu (S.M.D.); herzog.nora@gmail.com (N.L.H.); 4Rutgers New Jersey Medical School, Rutgers, The State University of New Jersey, Newark, NJ 07103, USA; alemenze@gmail.com (A.L.); donnelly@rutgers.edu (R.D.); 5Moffitt Cancer Center, Tampa, FL 33612, USA

**Keywords:** gut microbiota, machine learning, statistical algorithms, co-occurrence patterns, melanoma

## Abstract

Through a multitude of studies, the gut microbiota has been recognized as a significant influencer of both homeostasis and pathophysiology. Certain microbial taxa can even affect treatments such as cancer immunotherapies, including the immune checkpoint blockade. These taxa can impact such processes both individually as well as collectively through mechanisms from quorum sensing to metabolite production. Due to this overarching presence of the gut microbiota in many physiological processes distal to the GI tract, we hypothesized that mice bearing tumors at extraintestinal sites would display a distinct intestinal microbial signature from non-tumor-bearing mice, and that such a signature would involve taxa that collectively shift with tumor presence. Microbial OTUs were determined from 16S rRNA genes isolated from the fecal samples of C57BL/6 mice challenged with either B16-F10 melanoma cells or PBS control and analyzed using QIIME. Relative proportions of bacteria were determined for each mouse and, using machine-learning approaches, significantly altered taxa and co-occurrence patterns between tumor- and non-tumor-bearing mice were found. Mice with a tumor had elevated proportions of *Ruminococcaceae*, *Peptococcaceae*.g_rc4.4, and *Christensenellaceae,* as well as significant information gains and ReliefF weights for *Bacteroidales.f__S24.7*, *Ruminococcaceae*, *Clostridiales*, and *Erysipelotrichaceae*. *Bacteroidales.f__S24.7*, *Ruminococcaceae*, and *Clostridiales* were also implicated through shifting co-occurrences and PCA values. Using these seven taxa as a melanoma signature, a neural network reached an 80% tumor detection accuracy in a 10-fold stratified random sampling validation. These results indicated gut microbial proportions as a biosensor for tumor detection, and that shifting co-occurrences could be used to reveal relevant taxa.

## 1. Introduction

The gastrointestinal microbiota contains a diverse and dense collection of symbiotic organisms that contribute to intestinal homeostasis. Nutrient digestion, synthesis of vitamins, protection against pathologic organisms, and production of neurotransmitters are just a few of the biological functions that these organisms provide [1,2,3]. The host’s immune system plays an essential role in controlling microbial growth and development in the microbiome to ensure that a mutual relationship is maintained between the host and organism.

At the same time, the microbiota plays a role in adapting the host’s immune system to various stressors [4]. In fact, evidence is accumulating that the intestinal microflora can respond to changes in host health status by sensing soluble host elements and local micro-environmental cues [5]. For this reason, the gastrointestinal microbiota is affected by the pathological immune responses derived from diseases such as diabetes mellitus, cancer, obesity, and inflammatory diseases, which impacts the body’s immune response against disease [2,6,7].

It is increasingly being recognized that the gut microbiome composition differs significantly between healthy individuals and those with various pathological conditions. Dongmei et al. found that healthy individuals have a more diverse gut flora than those with colorectal cancer. In addition, certain bacterial populations were more likely to co-occur in patients with colorectal cancer than in healthy individuals [3]. While alterations in microbiome composition can be seen in pathologic conditions such as cancer, it is unclear whether these changes are a cause or a consequence of the disease [6]. Multiple studies that analyzed the composition of the gut microbiota in colorectal cancer patients suggested the presence of both “driver bacteria”, or those that promote cancer growth, and “passenger bacteria”, or those that solely flourish in the proinflammatory environment, but do not impact tumor progression. Geng et al. found that in their colorectal cancer patients, members of the *Enterobacteriaceae* family promoted cancer growth, whereas members of the *Streptococcaceae* family merely flourished in a proinflammatory environment [7].

The presence of these microbial mechanisms in which bacterial taxa have a certain level of dependency have wide implications for their use in modeling respective pathological conditions. Typically, connectivity and dependency between variables such as bacterial taxa in the context of predictive modeling has typically been a hindrance to model performance [8,9,10]. It is widely understood with many kinds of algorithms that, in various circumstances, variables with some manner of co-occurrence provide a certain level of redundant information, and therefore reduce the variability explained in models [8]. This presence of redundant information decreases the model’s fit to the training dataset, as well as its prediction accuracy in the testing dataset [10,11,12].

Despite these limitations, co-occurrences in the context of pathological prediction with microbial taxa may still hold significance in the application of diagnostic signatures [8,13]. When co-occurrences shift between conditions, so does the direction of variability represented by relevant taxa in planes of higher dimensionality [9,10,14]. These shifts are reflected in principal component analysis, in which each principal component represents a different proportion of the total variability present [8,13]. They are also represented in ReliefF and information gain values, in which microbial taxa with these differences in variability have increased reliability as predictors [11,15]. Therefore, the identification of these shifts in co-occurrences in pathological conditions such as cancer is optimal for the implementation of gut microbial diagnostic signatures.

The implementation of machine-learning algorithms for the prediction of the presence of various cancers using the gut microbiome has been widely studied [16,17,18]. However, to date, relatively little work has been done regarding the use of the gut microbiome to predict the presence of melanoma. In addition, one of the challenges of predicting the presence of a specific disease with the gut microbiota is the variability in relative proportions of specific gut bacteria that can exist between patients and populations [12]. Through our analyses, we have indicated shifts in microbial co-occurrences as a potential method in accounting for such variability. Therefore, we hypothesized that models based on gut microbial proportion profiles of taxa involved in co-occurrence shifts could form a distinct diagnostic signature that effectively differentiated mice bearing mouse melanoma tumors from non-tumor-bearing mice. This implies that the intestinal microflora may function as a biosensor for the presence of cancer, and that its manipulation may alter cancer prognoses.

## 2. Results

### 2.1. Shifts in Microbial Taxon Proportions of Melanoma-Bearing Mice

Mice bearing melanoma tumors displayed significant shifts in gut microbial proportions compared to non-tumor-bearing mice, which: (1) implicated consistency in changes in gut microbiota data with tumors in the skin, distal to the gut; and (2) implied that such changes could be used by an algorithm to detect the presence of cancer. We compared the microbial composition of fecal samples of melanoma-bearing and tumor-free mice by terminal restriction fragment length polymorphism (T-RFLP) analysis [14,16]. This technique is commonly used to study complex microbial communities based on 16S rRNA gene variation, and has been applied in the study of microbial communities in soil and sludge systems [19]. T-RFLP analysis was carried out in a blinded fashion as previously described [4]. It was readily seen for the two mouse experiments (Figure 1) that the co-occurrences of relative taxon proportions shifted in the presence of B16 melanoma. In addition, *Peptococcaceae*.g_rc4.4 was significantly increased (Wilcoxon *p* < 0.05) in both groups of mice (Figure 1). These data demonstrated that the intestinal flora developed detectable changes that discriminated a tumor-bearing from a tumor-free host. In order to more fully determine the extent to which these results distinguished between hosts that had a tumor and those that did not, the two mouse groups were combined and further analyzed as a single dataset (*n* = 56).

### 2.2. Co-Occurrence between Bacteroidales.f__S24.7, Clostridiales, and Ruminococcaceae Proportions in Mouse Melanoma

Seeking to identify the specific bacterial co-occurrences that were altered in the presence of a tumor, we first used Cytoscape to map them in the B16-melanoma- and PBS-treated mice. From these diagrams (Figure 2A,B), it was found that the co-occurrences of *Bacteroidales.f__S24.7* greatly differed between the two treatments. When looking further into this taxon, it was found that its co-occurrences with *Clostridiales* and *Ruminococcaceae* had changed the most between tumor and nontumor/PBS (Figure 2C,D), with Pearson correlation values of approximately −0.9 and −0.8 for tumor, as well as −0.15 and −0.13 for nontumor, respectively. Interestingly, however, when looking at the individual relative amounts of these taxa, the only one that was significantly different between tumor and nontumor was *Ruminococcaceae* (Wilcoxon *p* < 0.05, *T*-test *p* < 0.05; Figure 2E). Thus, we concluded that the potential for these taxa to predict tumor presence relied heavily on the extent to which their co-occurrences shifted in that condition, rather than changes in their individual relative amounts.

### 2.3. Differences in Principal Components between Tumor and Nontumor

Considering our results for both individual microbial taxa and co-occurrence shifts, we wanted to assess the relevance of each taxon in the context of predictive modeling. Thus, we calculated the information gains and ReliefF weights for each taxon (Figure 3A,B). In the scoring for information gains, *Ruminococcaceae*, *Peptococcaceae.g_rc4.4*, and *Christensenellaceae* consistently scored higher than the majority of taxa (Figure 3A). For the ReliefF algorithm, *Bacteroidales.f__S24.7* had a fairly high weight, along with *Peptococcaceae.g_rc4.4* and *Christensenellaceae* (Figure 3A). Further, *Christensenellaceae* was found to be significantly different between tumor and nontumor (Wilcoxon *p* < 0.05, Figure 3A,B). Considering that *Bacteroidales.f__S24.7* shifted its co-occurrences and its ReliefF weight indicated variable importance, we performed a principal component analysis (PCA) using this taxon (Figure 3C,D). Two PCAs were performed, one with *Clostridiales* and the other with *Ruminococcaceae* (Figure 3C,D). After performing the PCAs, we compared the resulting principal component coordinates between tumor and nontumor mice. From this comparison, we found that, although the first principal components did not differ between the two groups (Figure 3C), the second ones did (Wilcoxon *p* < 0.05, *T*-test *p* < 0.05; Figure 3D). These results indicated that the coordinates of these second principal components could be implemented in predictive modeling.

### 2.4. Prediction of Tumor Presence Using Microbial Taxa Involved in Altered Co-Occurrences

Since the second principal components involving *Bacteroidales.f__S24.7*, *Ruminococcaceae*, and *Clostridiales* were found to significantly differ with tumor presence, the proportions of those taxa, along with those of *Peptococcaceae.g_rc4.4*, *Christensenellaceae*, and *Erysipelotrichaceae,* were implemented as a mouse melanoma signature (Figure 4A,B). The 10-fold stratified random sampling used to obtain melanoma prediction results with machine-learning algorithms was performed by randomly selecting 90% of the mouse samples to train the algorithms and then testing them with the remaining 10% of samples (Figure 4A). This process was repeated 10 times, and the prediction results were averaged over those repeats (Figure 4A). Using this protocol, the highest percent accuracy in melanoma prediction was achieved by the neural network, with 80% (Figure 4A,B). Thus, the implementation of microbial taxa indicated by the second principal components in the prediction signature allowed for the identification of melanoma presence.

## 3. Discussion

Our findings demonstrated that the presence of a mouse melanoma tumor can be detected through the altered gut microbial proportions using classification algorithms. By using the gut microbial taxa to model tumor presence, it became apparent that such a condition manifested in more ways than just changes in individual amounts of certain taxa. Indeed, one of the main implications of this study is that considering gut microbial taxa co-occurrences and dependencies in predictive modeling can significantly increase predictive power in melanoma, more so than analyzing only statistical significance between groups. This concept of intertaxa correlations in modeling microbial-based conditions has wide applications in the interpretation of the gut microbiota, as it suggests that the role of an individual taxon in manifesting a biological phenotype is not solely attributed to its unique characteristics [17,18]. Rather, this role also depends on the extent to which a single taxon can communicate and affect other taxa through various mechanisms, from quorum sensing to metabolite production [20,21,22,23].

Despite this apparent, predictive relationship between murine melanoma and the gut microbiota, certain experimental limitations still existed. The primary limitation for consideration was the external validity of these results. It is often the case that gut microbiota data do not directly correspond between murine and human subjects, with various mechanisms implicated, from general differences in GI physiology to lifestyle, epigenetics, and immune responses [24,25,26]. Thus, in order for gut microbial associations to be implemented in clinical cancer diagnoses, further work needs to be done to elucidate pertinent taxa in a variety of human populations and pathophysiological states, including cancer, as well as the interaction between shifts in gut microbial content and certain factors such as diet and lifestyle. Most pertinent to patient treatment is the level of interaction between host immune responses and the gut microbiota, as antitumor immunity and immunotherapies may affect prediction outcomes [27,28]. These studies would also need to consider the correlation between patient stool sampling and gut microbial content with cancer presence, as sampling variation may be a confound [24]. Finally, since our gut microbiota data had a certain level of variation, other parameters should be considered in the future predictive modeling of human melanoma, such as biochemical and clinical observations [29].

In the statistical analysis of gut microbial taxa, algorithms have been developed to accurately detect the presence of these intertaxa co-occurrences [30,31,32]. Such algorithms for the detection of microbial “co-occurrence networks” include Sparse Inverse Covariance Estimation for Ecological Association Inference (SPEIC-EASI) and Sparse Correlations for Compositional Data (SparCC) [31,32,33]. However, despite these advances in the statistical detection of these interactions, there has not been as much work to determine their efficacy in different types of classification algorithms in conditions such as melanoma. In fact, their presence in predictive models has generally been discouraged, as the collinearity they create have been shown to compromise the performance of many model types [34,35,36]. Further, even for models that can more readily account for collinearity, the use of such interactions in these models does not consistently increase the performance of those models [34,35,36]. Thus, there is a necessity for a new statistical interpretation of intertaxa co-occurrences in order for them to be optimally utilized in a predictive model. Perhaps new insights into such interpretations can be eventually made when taxa indicated by shifts in co-occurrence networks are further tested in more architecturally complex algorithms such as deep-learning neural networks.

Traditionally, one of the most common procedures in dealing with collinearity between variables such as microbial taxa is the use of principal components in principal component analysis (PCA) [34,35,36,37]. By definition, the resulting principal components do not significantly correlate with each other, and are thus used in various model types [34,35,36,37]. These components are not usually interpretable from the perspective of the original data because they are linear transformations of that data [34,35,36,37]. However, if a small number of variables (e.g., two or three) is used, the principal components can be more easily interpreted [34,35,36,37]. In this study, PCA analysis was able to differentiate the two groups of mice successfully; however, much work still needs to be done to characterize the significance of individual PCs in different situations, such as in other clinically relevant tumor types.

## 4. Methods

### 4.1. Cell Culture

B16-F10 cells (ATCC) were cultured in RPMI 1640 plus 10% heat-inactivated fetal bovine serum (Atlanta Biologicals, Flowery Branch, GA, USA), 2 mM L-glutamine (Mediatech, Manassas, VA, USA), and 1% penicillin/streptomycin (Mediatech).

### 4.2. Mouse Experiments

C57BL/6 mice (B6; no. 00664; Jackson Laboratory) were housed in a specific pathogen-free facility at the Rutgers Cancer Institute of New Jersey. Experiments involving animals were carried out in accordance with respective Institutional Animal Care and Use Committee (IACUC) and Institutional Biosafety Committee (IBC) guidelines.

In the first experiment, 35 B6 male mice, aged 6 to 8 weeks old from the Jackson Laboratory were intradermally challenged in the right flank with 10^5^ cells of the highly aggressive and poorly immunogenic melanoma B16 cell line (*n* = 19) [17] or phosphate buffered saline (PBS) (*n* = 16) under isoflurane anesthesia. Mice were fed regular chow according to animal care institutional guidelines. Fecal sample collection to compare tumor-bearing to non-tumor-bearing mice was carried out on day 10, when tumors were approximately 25–50 mm^2^. Samples were stored immediately at −80 °C until DNA extraction [38] and sequencing.

The second experiment at this facility followed the identical protocol, using 21 B6 male mice aged 6 to 8 weeks old that were intradermally challenged in the right flank with 10^5^ cells of the highly aggressive and poorly immunogenic melanoma B16 cell line (*n* = 11) [17] or phosphate buffered saline (PBS) (*n* = 10) under isoflurane anesthesia. Fecal sample collection to compare tumor-bearing to non-tumor-bearing mice was carried out on day 16, when tumors were approximately 25–50 mm^2^ in diameter. Samples were stored immediately at −80 °C until DNA extraction [38] and sequencing.

### 4.3. DNA Extraction

Fecal pellets were homogenized and extracted using the QIAamp PowerFecal DNA Extraction kit following the manufacturer’s protocols [39].

### 4.4. 16S rRNA Gene Sequencing and Data Analysis

The 16S rRNA genes were amplified from purified DNA using PCR primers specific to the V3–V4 region of the 16S rRNA gene and sequenced by Illumina MiSeq in a 2 × 150 bp configuration at the Rutgers New Jersey Medical School Genomics Core. Quantitative Insights Into Microbial Ecology (QIIME) software was used for open-reference operational taxonomic unit (OTU) classification with OTU clustering at 0.97, followed by rarefaction and taxonomic classification of de novo OTUs [40].

### 4.5. qPCR for Bacterial Load and Taxa Assays

Bacterial loads of extracted fecal DNA were determined by qPCR. DNA were quantified against a standard curve, and the results were normalized to the weight of fecal samples [40].

### 4.6. Taxon Comparisons, Analyses, and Statistical Modeling

Using the R programming language, microbial taxa between tumor-bearing and PBS control mice were compared using Welch’s *t*-test as well as the Mann–Whitney *U* test (a *p*-value of <0.05 was considered to denote statistically significant differences). Between these two groups of mice, general taxa and comparison attributes were determined using the Orange3 v3.27.1 data-mining program and the CORElearn package in CRAN. PCA analysis and principal components were determined using the prcomp function in R. General machine-learning model analyses and cross-validation procedures were performed using the Orange3 program with these settings:

The neural network was a 100-neuron single hidden layer that used the ReLu activation function and the Adam solver.

The support vector machine (SVM) used a radial basis function (RBF) kernel with a cost of 1.0 and a regression loss epsilon of 0.1.

The AdaBoost used a SAMME.R classification algorithm with a linear regression loss function, 50 estimators, and learning rate of 1.0.

The CN2 rule inducer used entropy as the evaluation measure, a beam width of 5, and a maximum rule length of 5.

The random forest used a 12-tree ensemble with subsets split no smaller than 5.

The k-nearest neighbor (kNN) used 5 neighbors and considered the Euclidean distance and uniform weights.

For the naïve Bayes, the attributes were not weighted.

Tree used a maximal tree depth of 100 and subsets not split smaller than 5.

In the logistic regression, a ridge regularization was implemented.

Quality parameters for this model were determined using an internal 10-fold stratified shuffle split, with 90% of the samples selected for training and the remaining 10% for testing in Orange3. Results were graphed using the ggplot2, ggrepel, and ggpubr packages in CRAN, as well as Orange3 and Cytoscape v3.7.2. Heatmaps were generated using the ComplexHeatmap package in CRAN. Tables were formatted using the sjPlot package in CRAN.

## Figures and Tables

**Figure 1 diagnostics-12-00958-f001:**
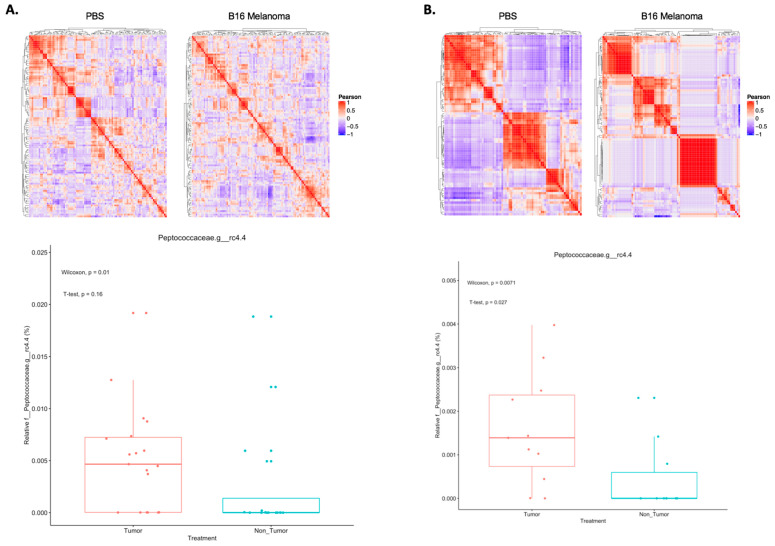
Shifted co-occurrences of microbial taxa and increased *Peptococcaceae*.g_rc4.4 characterize tumor presence. (**A**) C57BL/6 (B6) male mice were injected with either 10^5^ B16 melanoma cells (*n* = 19) or PBS (*n* = 16). After 10 days, fecal samples were collected and 16S rRNA genes were analyzed using terminal restriction fragment length polymorphism (T-RFLP) analysis. From individual taxon proportion and co-occurrence patterns, it could be seen that such patterns shifted with melanoma presence, and *Peptococcaceae*.g_rc4.4 levels increased. (**B**) B6 male mice were injected with either 10^5^ B16 melanoma cells (*n* = 11) or PBS (*n* = 10). After 16 days, fecal samples were collected and 16S rRNA genes were analyzed using terminal restriction fragment length polymorphism (T-RFLP) analysis. The results of these data directly corresponded with the mice in (**A**).

**Figure 2 diagnostics-12-00958-f002:**
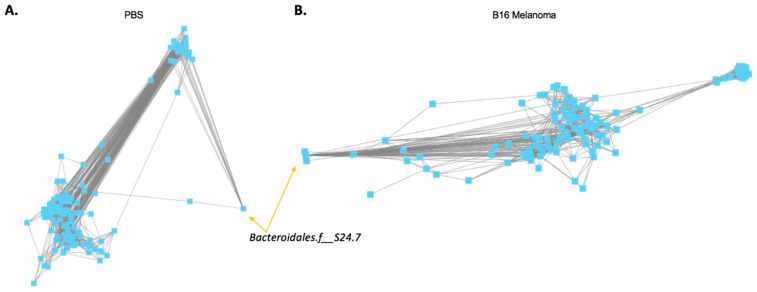
Co-occurrence changes between *Bacteroidales.f__S24.7*, *Clostridiales,* and *Ruminococcaceae* occur with tumor presence. (**A**,**B**) Pearson correlation matrices were determined for microbiotas from tumor and nontumor mice and displayed using Cytoscape. From these visualizations, *Bacteroidales.f__S24.7* co-occurrences greatly changed with tumor presence. (**C**,**D**) Using the R programming language, it was found that the most dramatic shifts of *Bacteroidales.f__S24.7* were in conjunction with *Clostridiales* and *Ruminococcaceae*. (**E**) When comparing each taxon individually between tumor and nontumor, only *Ruminococcaceae* was significantly different.

**Figure 3 diagnostics-12-00958-f003:**
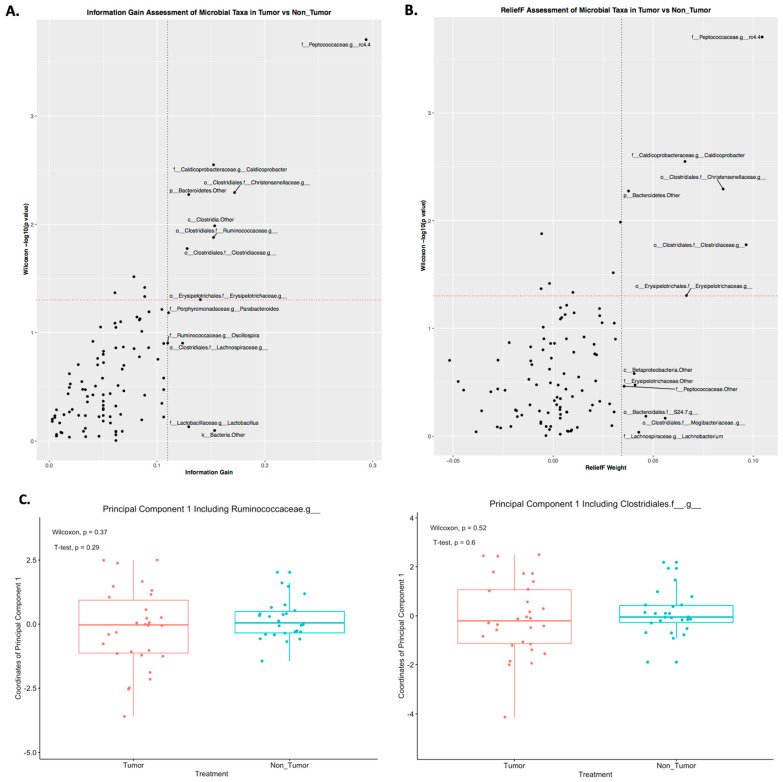
Significant predictors of tumor presence include the second principal components involving *Bacteroidales.f__S24.7*, *Clostridiales,* and *Ruminococcaceae*. (**A**,**B**) Using the CORElearn package in the R programming language, the information gains and ReliefF weights were calculated for each taxon. (**A**) *Ruminococcaceae*, *Peptococcaceae.g_rc4.4*, and *Christensenellaceae* were found significantly altered with tumor presence and having high information gains. (**B**) Along with *Peptococcaceae.g_rc4.4* and Christensenellaceae, *Bacteroidales.f__S24.7* and *Erysipelotrichaceae* had high ReliefF weights. (**C**,**D**) Two PCAs using *Bacteroidales.f__S24.7*, one with *Ruminococcaceae* and the other with *Clostridiales*, were conducted using R. While their first principal components did not change with tumor, their second ones did (Wilcoxon *p* < 0.05, *T*-test *p* < 0.05 (**D**)).

**Figure 4 diagnostics-12-00958-f004:**
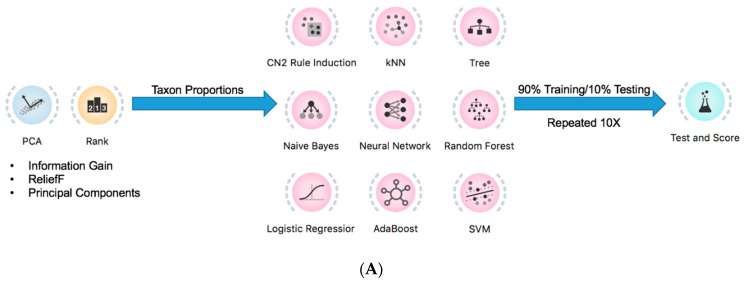
Implementation of microbial taxa implicated in second principal components accurately predict tumor presence. (**A**) Using Orange3, 10-fold stratified shuffle splits were performed. (**B**) Using a prediction signature which included *Bacteroidales.f__S24.7*, *Ruminococcaceae*, and *Clostridiales*, implicated in the second principal components, resulted in an average accuracy of 80% achieved with a Neural Network classifier. AUC, area under the curve; CA, classification accuracy; F1, F1 score).

## Data Availability

The data supporting the findings of this study are available from the corresponding authors upon reasonable request.

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
