# Peer review of "Gut Microbial Shifts Indicate Melanoma Presence and Bacterial Interactions in a Murine Model"

_diagnostics, 2022, doi:10.3390/diagnostics12040958_

Round 1
Reviewer 1 Report
The authors present a very interesting manuscript based on harnessing melanoma murine models to unravel bacterial gut microbiota profiles. To accomplish this, added to a sound hypothesis and experimental framework, they used appropriate machine learning to dissect such discriminatory patterns. Very good that they acknowledge the limitations of such methodology, in particular unsupervised methods such as principal component analysis. They have justified their dimensionality reduction with experimental context and provide a set of trained methods that can differentiate the tumour classes with accuracies of upto 0.80 with overall good precision and recall.
I suggest only minor revision to make the manuscript more readable and succinct as follows
Minor Corrections:
Line 26:
Where: Relative proportions of bacteria were determined for each mouse and, using machine learning and statistical algorithms in Orange3 and the R programming language, significantly altered taxa and co-occurrence patterns between tumor- and non-tumor-bearing mice were found.
Change to: Relative proportions of bacteria were determined for each mouse and, using machine learning approaches, significantly altered taxa and co-occurrence patterns between tumor- and non-tumor-bearing mice were found.
Line 33
Where: Using these seven taxa as a melanoma signature, a Neural Network reached an 80% tumor detection accuracy in a 10-fold stratified random sampling validation. These results indicate gut microbial proportions as a biosensor for tumor detection and that shifting cooccurrences and PCA values reveal relevant taxa.
Change to: Using these seven taxa as a melanoma signature, a Neural Network reached an 80% tumor detection accuracy in a 10-fold stratified random sampling validation. These results indicate gut microbial proportions as a biosensor for tumor detection and that shifting cooccurrences could be used to reveal relevant taxa.
Line 238
Where: In the case of two variables, or one variable correlating with another, the second principal component (PC) is more likely to differentiate between more heavily correlated data points and those that are more dispersed because it is perpendicular to the first PC. This interpretability of the principal components in low numbers of variables allows for a more effective implementation of PCA in cancer signature identification.
Suggestion to the authors: please consider to simplify the argument by removing this paragraph.
Line 242:
Where: Although, for this study, the second PC was able to differentiate the two groups of mice successfully, much work still needs to be conducted to characterize the significance of individual PCs in different situations, such as other disease types and greater numbers of variables.
Change to: Although, for this study, PCA analysis was able to differentiate the two groups of mice successfully, much work still needs to be conducted to characterize the significance of individual PCs in different situations, such as in other clinically relevant tumour types
Author Response
Manuscript ID: diagnostics-1605925
Title: "Gut microbial shifts indicate melanoma presence and bacterial interactions in a murine model"
Diagnostics
We would like to thank Reviewer 1 for taking the time to review our manuscript in this peer review process. We were pleased to read the reviewer’s positive remarks, including that we presented “a very interesting manuscript” and “a sound hypothesis,” used “appropriate machine learning,” acknowledge “limitations of such methodology,” and can differentiate tumor classes with “overall good precision and recall.” We also appreciate the thoughtful comments that Reviewer 1 provided. We have addressed these in our point-by-point responses below. We believe that these revisions indeed improve the readability and succinctness of our manuscript. We are submitting a marked (with yellow highlight) version of the newly revised manuscript.
REVIEWER 1 COMMENTS
Comment 1) Line 26: Where: Relative proportions of bacteria were determined for each mouse and, using machine learning and statistical algorithms in Orange3 and the R programming language, significantly altered taxa and co-occurrence patterns between tumor- and non-tumor-bearing mice were found. Change to: Relative proportions of bacteria were determined for each mouse and, using machine learning approaches, significantly altered taxa and co-occurrence patterns between tumor- and non-tumor- bearing mice were found.
Response 1: Thank you for the revision. We have made this change in our edited manuscript.
Comment 2) Line 33: Where: Using these seven taxa as a melanoma signature, a Neural Network reached an 80% tumor detection accuracy in a 10-fold stratified random sampling validation. These results indicate gut microbial proportions as a biosensor for tumor detection and that shifting cooccurrences and PCA values reveal relevant taxa. Change to: Using these seven taxa as a melanoma signature, a Neural Network reached an 80% tumor detection accuracy in a 10- fold stratified random sampling validation. These results indicate gut microbial proportions as a biosensor for tumor detection and that shifting cooccurrences could be used to reveal relevant taxa.
Response 2: Thank you for the revision. We have made this change in our edited manuscript.
Comment 3) Line 238: Where: In the case of two variables, or one variable correlating with another, the second principal component (PC) is more likely to differentiate between more heavily correlated data points and those that are more dispersed because it is perpendicular to the first PC. This interpretability of the principal components in low numbers of variables allows for a more effective implementation of PCA in cancer signature identification. Suggestion to the authors: please consider to simplify the argument by removing this paragraph.
Response 3: Thank you for the revision. We have made this change in our edited manuscript.
Comment 4) Line 242: Where: Although, for this study, the second PC was able to differentiate the two groups of mice successfully, much work still needs to be conducted to characterize the significance of individual PCs in different situations, such as other disease types and greater numbers of variables. Change to: Although, for this study, PCA analysis was able to differentiate the two groups of mice successfully, much work still needs to be conducted to characterize the significance of individual PCs in different situations, such as in other clinically relevant tumour types
Response 4: Thank you for the revision. We have made this change in our edited manuscript.
Reviewer 2 Report
The authors present an interesting application of gut microbiota assessment.
However, there are some very important limits.
- The intestinal physiology and therefore the mycobiota of rats is very different from that of humans, so the results could be very different.
- The authors rightly point out the influence of lifestyle and nutrition on the microbiota but these aspects are scarcely considered in the experiment.
- Should it be clarified, even if only by hypothesis, why the mycorbiota has a variation of this type, in order to clarify what could be false positives, that is, is this situation an exclusive peculiarity of a tumor?
- The data, showing a statistically significant variation, are very scattered with values very far from each other in some cases, thus showing an individual variability that could affect the goodness of the possible marker
- Finally, thinking about its use in humans also how the feces are produced and what consistency they have is a variable to consider
Author Response
Manuscript ID: diagnostics-1605925
Title: "Gut microbial shifts indicate melanoma presence and bacterial interactions in a murine model"
Diagnostics
We would like to thank Reviewer 2 for taking the time to review our manuscript in this peer review process. We were pleased to read that we, “present an interesting application of gut microbiota assessment”. We also appreciate the thoughtful comments that you provided, specifically regarding limitations that deserve additional mention in our text. We have responded to the comments in our point-by-point responses below. We believe that these revisions improve the clarity and impact of our manuscript. We are submitting a marked (with yellow highlight) version of the newly revised manuscript with resultant new text and additional citations.
REVIEWER 2 COMMENTS
Comment 1) The intestinal physiology and therefore the mycobiota of rats is very different from that of humans, so the results could be very different.
Response 1: Indeed, the degree to which this study applies to human populations is a limitation that may be attributed to a variety of factors, from differences in general physiology to lifestyle and individual gene expression. We address this limitation in the revised version in lines 3-6 and 12-14 of the added Discussion paragraph.
Comment 2) The authors rightly point out the influence of lifestyle and nutrition on the microbiota but these aspects are scarcely considered in the experiment.
Response 2: In our experimental design, we decided to focus on the presence of a B16 melanoma tumor as an initial study. We do, however, intend to investigate the impact of these factors on the corresponding gut microbiota in the future. We address this in the revised version in lines 6-10 of the added Discussion paragraph.
Comment 3) Should it be clarified, even if only by hypothesis, why the mycorbiota has a variation of this type, in order to clarify what could be false positives, that is, is this situation an exclusive peculiarity of a tumor?
Response 3: This situation of variability with regard to gut microbiota data is not unique to tumor biology. In fact, it is found in a plethora of conditions, from Intestinal Bowel Disease (IBD) to Parkinson’s Disease. This variability is attributed to the connection between the gut microbiota and many physiological systems. With regard to our specific study with B16 melanoma, one of our hypotheses is that individual differences in tumor immune responses contribute to variability in our gut microbiota data, as multiple studies have indicated a correspondence between the gut microbiota and anti-tumor immunity. We address this in he revised version in lines 10-12 of the added Discussion paragraph.
Comment 4) The data, showing a statistically significant variation, are very scattered with values very far from each other in some cases, thus showing an individual variability that could affect the goodness of the possible marker
Response 4: Indeed, although statistically significant, there is variability in our gut microbiota data. This is a limitation that we address in our revised manuscript lines 14-16 of the added Discussion paragraph. These results also instigate further studies into the implementation of other pathophysiological variables in addition to the gut microbiota, so as to further increase the accuracy of tumor detection.
Comment 5) Finally, thinking about its use in humans also how the feces are produced and what consistency they have is a variable to consider
Response 5: Indeed, human patient sampling and variability with regard those samples are factors to consider for the applicability of this study in the clinical setting. These factors are considered in our revised manuscript lines 12-14 of the added Discussion paragraph.
Round 2
Reviewer 2 Report
The authors have made some important changes, therefore, although preliminary, the manuscript can be accepted